# Design, Synthesis, and In Vitro Evaluation of Benzofuro[3,2-*c*]Quinoline Derivatives as Potential Antileukemia Agents

**DOI:** 10.3390/molecules25010203

**Published:** 2020-01-03

**Authors:** Ying Lin, Dong Xing, Wen-Biao Wu, Gao-Ya Xu, Li-Fang Yu, Jie Tang, Yu-Bo Zhou, Jia Li, Fan Yang

**Affiliations:** 1Shanghai Engineering Research Center of Molecular Therapeutics and New Drug Development, School of Chemistry and Molecular Engineering, East China Normal University, Shanghai 200062, China; shanghailinying902@163.com (Y.L.); dxing@sat.ecnu.edu.cn (D.X.); lfyu@sat.ecnu.edu.cn (L.-F.Y.); jtang@chem.ecnu.edu.cn (J.T.); 2National Center for Drug Screening, Shanghai Institute of Material Medica, Chinese Academy of Science, Shanghai 201203, China; s18-wuwenbiao@simm.ac.cn (W.-B.W.); gyxu@simm.ac.cn (G.-Y.X.); ybzhou@simm.ac.cn (Y.-B.Z.); 3University of Chinese Academy of Sciences, No. 19A Yuquan Road, Beijing 100049, China; 4Shanghai Greenchem & Biotech Co. Ltd., Shanghai 200062, China; 5Open Studio for Druggability Research of Marine Natural Products, Pilot National Laboratory for Marine Science and Technology (Qingdao), 1 Wenhai Road, Aoshanwei, Jimo, Qingdao 266237, China

**Keywords:** benzofuro[3,2-*c*]quinolines, 3-(2-methoxyphenyl)quinolin-4(1*H*)one, MV-4-11 cell line, antileukemia activity

## Abstract

Herein, we design and synthesize an array of benzofuro[3,2-*c*]quinolines starting from 3-(2-methoxyphenyl)quinolin-4(1*H*)ones via a sequential chlorination/demethylation, intramolecular cyclization pathway. This sequential transformation was efficient, conducted under metal-free and mild reaction conditions, and yielded corresponding benzofuro[3,2-*c*]quinolines in high yields. In vitro biological evaluation indicated that such type of compounds showed excellent antileukemia activity and selectivity, and therefore may offer a promising hit compound for developing antileukemia compounds.

## 1. Introduction

Furo[3,2-*c*]pyridines belong to a novel class of heterocycles that exhibit a wide range of biological activities [1,2,3]. A large number of heterocyclic compounds bearing a furo[3,2-*c*]pyridine core have been reported as anticancer (Figure 1A, compounds **A** and **B**) [4,5] or antibacterial agents (Figure 1A, compound **C**) [6]. Among them, the tricyclic furo[3,2-*c*]quinoline skeleton possessing a fused aryl ring on the pyridine part has also shown unique pharmaceutical and biological activities. For example, compound **D** has been found to be a neuroprotective agent [7]. Compound **E** was reported to be a potent antiproliferative molecule against renal UO-31, melanoma UACC-257, and UACC-62 cell lines [8]. Agrawal et al. reported that compound **F** may be a potential topoisomerase-II inhibitor based on related computational studies [9]. In view of the diverse biological activities of such molecules, the design and synthesis of novel heterocyclic structures possessing the furo[3,2-*c*]pyridine skeleton would be highly valuable for biological studies. As our continuous research interest, herein we design and synthesize a new type of tetracyclic benzofuro[3,2-*c*]quinoline (Figure 1B, compound **G**) that contains the key furo[3,2-*c*]pyridine skeleton by introducing a fused aryl ring on the furan side. Although such types of tetracyclic heterocycles have been reported by several research groups, related biological studies are still very limited [10]. Based on our newly synthesized compounds, an in vitro biological evaluation was then conducted. 

Recently, our group has developed an efficient route for the synthesis of 3-(2-methoxyphenyl)quinolin-4(1*H*)-ones 1 via an in situ Meinwald rearrangement/intramolecular reductive cyclization of chalcone epoxides (Figure 1B) [11,12]. This transformation was easily scaled up, showed a broad substrate scope, and tolerated a variety of functional groups. With this method developed, we envisioned that an efficient synthesis of benzofuro[3,2-*c*]quinoline **G** could be designed via demethylation followed by either one-pot or stepwise cyclization starting from 3-(2-methoxyphenyl)quinolin-4(1*H*)-ones 1 (Figure 1B). A batch of methods for the synthesis of such compounds have been reported, such as the palladium-catalyzed aromatization reaction [10], O-arylation/dehydrogenative cross-coupling reaction [13], C-O coupling reaction [14], demethyl-cyclization reaction [15], and many others [16,17,18]. The current method is featured due to it using a readily accessible starting material and involving the easy installation of different functional groups and the mild and metal-free reaction conditions to access a number of substituted benzofuro[3,2-*c*]quinoline derivatives for in vitro biological evaluations. Furthermore, it was found that this type of compound showed promising antileukemia activities.

## 2. Results and Discussion

### 2.1. Chemistry

Initially, 3-(2-methoxyphenyl)quinolin-4(1*H*)-one 1a, which was conveniently obtained from the corresponding chalcone epoxide, was chosen as the model substrate for the synthesis of benzofuro[3,2-*c*]quinoline **2a** via a demethylation/cyclization cascade. However, upon a thorough condition screening, the formation of **2a** could not be observed at all [Appendix A]. In order to access the desired product **2a** from the readily available starting substrate **1a**, a stepwise reaction was then developed (Scheme 1). First, by treating with suitable chlorination reagents, **1a** was converted into the corresponding 4-chloroquinine **3a**. Upon carefully investigating condition optimizations, SOCl_2_ was found to be the most effective, yielding **3a** with a 93% yield using CH_2_Cl_2_ as the solvent. Treating **3a** with 48% HBr under reflux gave the demethylation product **4a** with a 98% yield. The chlorination and demethylation steps were conducted in a one-pot manner without the isolation of **3a**, and the overall yield of **4a** was maintained at a high level (96%). Finally, a simple intramolecular annulation of **4a** with KO*t*-Bu led to the formation of the expected benzofuro[3,2-*c*]quinoline **2a** with a high yield [Appendix A].

With the optimized reaction conditions identified, the scope of this stepwise chlorination/demethylation/cyclization transformation was investigated by preparing different substituted benzofuro[3,2-*c*]quinolines (Table 1). Several substrates bearing different functional groups on both the benzofuran and the quinoline were prepared and subjected to the standard reaction conditions. Halogen substituents at different positions on the benzofuran ring were all well tolerated, giving the corresponding tetracyclic products in high overall yields (**2c**, **2f**–**2i**). A substrate bearing a trifluoromethoxyl substituent on the benzofuran ring also worked well (**2k**). With the methyl substituent at different positions on the benzofuran ring, this two-step transformation ran efficiently, providing the corresponding cyclization products in good overall yields (**2b**, **2d**, and **2j**). The substrate scope with different substituents on the quinoline moiety was then investigated. Both the electron-donating methyl substituent (**2n**, **2p**, **2t**, and **2v**) and electron-withdrawing halogen substituents (**2o** and **2r**–**2s**) at different positions all gave the corresponding cyclization products in good overall yields. A free hydroxyl substituent was introduced to different positions on both the benzofuran and quinoline rings, with the aim of increasing the potential biological activities of these molecules [19,20,21]. While the one-pot chlorination/demethylation step was still very efficient, the subsequent base-promoted cyclization for these hydroxyl-substituted substrates generally gave low yields of the corresponding cyclization products. The low yield of the second step may have come from the competitive reactions between the free hydroxyl groups with the base (**2e**, **2l**, **2q**, **2u**, and **2w**–**2x**), and we speculate that the low yields of the second steps may be improved via selective protection of the hydroxyl group, which is not involved in the cyclization reaction.

### 2.2. Benzofuro[3,2-c]Quinoline Inhibition Studies

With these newly synthesized benzofuro[3,2-c]quinoline derivatives available, their in vitro antileukemia activities were evaluated against the MV-4-11 cell line, which is one of the acute myelocytic leukemia (AML) cell lines (Table 2) [22,23]. As shown in Table 2, most of these compounds, except for **2h**, **2k**–**2m**, and **2v**, exhibited good antileukemia activity against MV-4-11 cells. Among them, compound **2e**, possessing a free hydroxyl group at the C5 position on the benzofuran ring, showed the highest antileukemia activity (IC_50_ = 0.12 μM). When the free hydroxyl group was replaced with Cl or CH_3_, the activities decreased 10-fold (**2d** and **2g**). Compound **2q**, bearing a free hydroxyl group at C8 on the quinoline ring, also showed promising antileukemia activity (IC_50_ = 0.24 μM). Similarly, when this hydroxyl substituent was replaced with Cl, the activity was reduce 10-fold (**2o**). On the other hand, replacing this hydroxyl group with methyl resulted in a slightly decreased activity (**2p**). Methyl, halogen, or hydroxyl substitutions at C3 showed only a slight difference on antileukemia activity (**2r**–**2u**). As concluded from these results, compounds with hydroxyl substitution (**2e**, **2q**, and **2u**) generally exhibited good activities; however, compounds bearing a free hydroxyl group at C4 of the benzofuran ring (**2l** and **2x**) did not lead to increased activities.

It is vital to examine the cytotoxicity on normal cells during the development of an anticancer drug. Accordingly, four compounds that showed high antileukemia activities—**2e**, **2p**, **2q**, and **2u**—were selected for the selectivity test on peripheral blood mononuclear cells (PBMCs) using an MTS assay (Table 3). The selectivity indexes (SIs) were counted by dividing the IC_50_ values in a PBMC by the IC_50_ values in MV-4-11 cell lines [25]. It was obvious that benzofuro[3,2-c]quinolines were less toxic on human peripheral blood mononuclear cell compared to human leukemia MV-4-11 cells. The most active compound, **2e** (SI = 79.5), showed a 79.5 times higher selectivity toward cancer cells than toward normal cells.

## 3. Materials and Methods

### 3.1. Chemical Synthesis

#### 3.1.1. General

All reagents were commercially available and used without purification. The progress of the reactions was monitored using silica gel thin-layer chromatography (TLC). Flash column chromatography was conducted using silica gel (Yantai Kangbinuo, ShanDong, Yantai, China, 200–300). ^1^H- and ^13^C-NMR spectra (Bruker Ascend, Bremen, Germany) were recorded at 400 and 100 MHz, respectively, using TMS (trimethyl chlorosilane) as the internal standard. Melting points were measured using a WRR-Y drug melting point measurement (Shanghai INESA Physico-Optical Instrument Co. Ltd, Shanghai, China). The high-resolution mass spectra were recorded on a Bruker micOTOF II spectrometer (Electrospray Ionization).

#### 3.1.2. General Procedures for the Synthesis of 2-(4-Chloroquinolin-3-Yl)Phenol Substrates **4a**–**4x**

A mixture of the substituted 3-phenylquinolin-4(1H)-one (1 equiv.), SOCl_2_ (4 equiv.), dichloromethane (5 mL) was refluxed until the reaction was completed (1 h), as evidenced using thin-layer chromatography (petroleum ether/ethyl acetate, 10:1). The cooled reaction mixture was concentrated in vacuo, diluted with water, then extracted with dichloromethane (3 × 20 mL). The combined organic layer were washed with water and brine, dried with anhydrous Na_2_SO_4_, and concentrated in vacuo. Then, the crude mixture was refluxed under 48% HBr without further purification until the reaction was completed. The mixture was poured into ice water and neutralized with saturated NaHCO_3_. The aqueous phase was extracted with ethyl acetate (3 × 20 mL), dried with anhydrous Na_2_SO_4_, and concentrated in vacuo. The crude mixture was purified using column chromatography over a silica gel using dichloromethane/methanol (80:1) as an eluent.

*2-(4-Chloroquinolin-3-yl)phenol* (**4a**): White solid (0.25 g, 96%), ^1^H-NMR (400 MHz, DMSO-*d*_6_) δ 9.76 (s, 1H), 8.78 (s, 1H), 8.29 (d, *J* = 8.3 Hz, 1H), 8.13 (d, *J* = 8.3 Hz, 1H), 7.89 (dd, *J* = 7.5, 4.1 Hz, 1H), 7.80 (dd, *J* = 7.8, 7.5 Hz, 1H), 7.34–7.28 (m, 2H), 7.02 (d, *J* = 8.1 Hz, 1H), 6.96 (dd, *J* = 7.4, 7.3 Hz, 1H). ^13^C-NMR (100 MHz, DMSO*-d*_6_) δ 154.9, 152.2, 147.3, 139.6, 131.2, 130.9, 130.2, 130.0, 129.3, 128.2, 125.6, 124.0, 122.8, 119.0, 115.8.

*2-(4-Chloroquinolin-3-yl)-6-methylphenol* (**4b**): Orange solid (0.21 g, 78%), ^1^H-NMR (400 MHz, CDCl_3_) δ 8.94 (s, 1H), 8.47–8.44 (m, 1H), 8.36 (d, *J* = 8.2 Hz, 1H), 8.04–8.01 (m, 1H), 7.92–7.88 (m, 1H), 7.15 (d, *J* = 4.6 Hz, 1H), 7.01 (d, *J* = 5.7 Hz, 1H), 6.84–6.81 (m, 1H), 2.18 (s, 3H). ^13^C-NMR (100 MHz, CDCl_3_) δ 156.5, 156.3, 149.8, 140.7, 139.2, 137.2, 137.0, 135.3, 132.4, 131.8, 130.2, 130.0, 125.1, 124.3, 124.0, 20.2.

*2-(4-Chloroquinolin-3-yl)-6-fluorophenol* (**4c**): Yellow solid (0.24 g, 87%), ^1^H-NMR (400 MHz, DMSO*-d*_6_) δ 9.94 (br s, 1H), 8.91 (s, 1H), 8.34 (d, *J* = 8.1 Hz, 1H), 8.18 (d, *J* = 8.3 Hz, 1H), 7.97 (dd, *J* = 7.6, 7.1 Hz, 1H), 7.86 (dd, *J* = 7.5, 7.4 Hz, 1H), 7.35–7.30 (m, 1H), 7.14 (d, *J* = 7.6 Hz, 1H), 6.97 (ddd, *J* = 8.0, 7.9, 5.0 Hz, 1H). ^13^C-NMR (100 MHz, DMSO*-d*_6_) δ 151.7 (d, J_C-F_ = 238.1 Hz), 150.9, 145.7, 142.7 (d, J_C-F_ = 14.6 Hz), 141.8, 131.3, 129.8 (d, J_C-F_ = 3.0 Hz), 128.9, 128.0, 126.6 (d, J_C-F_ = 2.8 Hz), 125.7, 125.5 (d, J_C-F_ = 3.3 Hz), 124.3, 119.3 (d, J_C-F_ = 7.4 Hz), 116.5 (d, J_C-F_ = 18.2 Hz).

*2-(4-Chloroquinolin-3-yl)-5-methylphenol* (**4d**): Yellow-green solid (0.22 g, 81%), ^1^H-NMR (400 MHz, DMSO*-d*_6_) δ 9.64 (br s, 1H), 8.76 (s, 1H), 8.28 (d, *J* = 8.2 Hz, 1H), 8.12 (d, *J* = 7.9 Hz, 1H), 7.88 (dd, *J* = 7.4, 7.2 Hz, 1H), 7.79 (dd *J* = 7.1, 7.0 Hz, 1H), 7.16 (d, *J* = 7.6 Hz, 1H), 6.82 (s, 1H), 6.77 (d, *J* = 7.5 Hz, 1H), 2.31 (s, 3H). ^13^C-NMR (100 MHz, DMSO*-d*_6_) δ 154.7, 152.3, 147.1, 139.6 (2C), 130.9, 130.2, 129.2, 128.2, 125.6, 124.0 (2C), 119.9, 119.8, 116.3, 21.0.

*4-(4-Chloroquinolin-3-yl)benzene-1,3-diol* (**4e**): Black oil (0.24 g, 90%), ^1^H-NMR (400 MHz, CD_3_OD/CDCl_3_ = 2/3) δ 8.89–8.88 (m, 1H), 8.38–8.36 (m, 2H), 7.96–7.92 (m, 1H), 7.87–7.81 (m, 1H), 7.02 (dd, *J* = 8.3, 6.8 Hz, 1H), 6.47–6.46 (m, 1H), 6.35 (dd, *J* = 6.8, 6.4 Hz, 1H). ^13^C-NMR (100 MHz, CD_3_OD/CDCl_3_ = 2/3) δ 157.4, 153.0, 148.7, 143.7, 134.1, 131.9, 129.9, 129.3, 128.4, 125.2, 123.2, 118.8, 108.2, 105.4, 100.4.

*5-Bromo-2-(4-chloroquinolin-3-yl)phenol* (**4f**): White solid (0.31g, 94%), ^1^H-NMR (400 MHz, CDCl_3_) δ 8.65 (s, 1H), 8.04 (d, *J* = 8.4 Hz, 1H), 7.94 (d, *J* = 8.4 Hz, 1H), 7.69 (dd, *J* = 8.1, 7.2 Hz, 1H), 7.51 (dd, *J* = 8.0, 7.2 Hz, 1H), 7.30 (s, 1H), 7.19–7.16 (m, 1H), 7.08 (d, *J* = 8.2 Hz, 1H). ^13^C-NMR (100 MHz, CDCl_3_) δ 154.2, 150.5, 145.9, 141.4, 131.4, 129.7, 128.2, 127.4, 127.2, 125.0, 123.5, 122.7, 122.5, 120.7, 119.5.

*5-Chloro-2-(4-chloroquinolin-3-yl)phenol* (**4g**): White solid (0.26 g, 88%), ^1^H-NMR (400 MHz, CD_3_OD/CDCl_3_ = 2/3) δ 8.70 (s, 1H), 8.32 (d, *J* = 8.4 Hz, 1H), 8.05 (d, *J* = 8.3 Hz, 1H), 7.78 (dd, *J* = 8.0, 7.1 Hz, 1H), 7.67 (dd, *J* = 8.1, 7.2 Hz, 1H), 7.18 (d, *J* = 8.1 Hz, 1H), 6.98–6.93 (m, 2H). ^13^C-NMR (100 MHz, CD_3_OD/CDCl_3_ = 2/3) δ 154.8, 150.8, 146.3, 141.0, 134.4, 131.4, 129.5, 129.4, 127.6, 127.1, 125.8, 123.8, 120.8, 118.8, 115.2.

*2-(4-Chloroquinolin-3-yl)-4-fluorophenol* (**4h**): Yellow solid (0.22 g, 80%), ^1^H-NMR (400 MHz, CD_3_OD/CDCl_3_ = 2/3) δ 8.73 (s, 1H), 8.33 (d, *J* = 8.3 Hz, 1H), 8.06 (d, *J* = 8.4 Hz, 1H), 7.79 (dd, *J* = 7.9, 7.3 Hz, 1H), 7.68 (dd, *J* = 8.0, 7.2 Hz, 1H), 7.01–6.97 (m, 2H), 6.93–6.90 (m, 1H). ^13^C-NMR (100 MHz, CD_3_OD/CDCl_3_ = 2/3) δ 155.2 (d, J_C-F_ = 235.9 Hz), 150.7, 150.1 (d, J_C-F_ = 1.8 Hz), 146.4, 141.0, 129.6, 129.3, 127.7, 127.2, 125.8, 123.8, 122.8 (d, J_C-F_ = 8.0 Hz), 116.6 (d, J_C-F_ = 23.4 Hz), 115.8 (d, J_C-F_ = 8.0 Hz), 115.6 (d, J_C-F_ = 22.7 Hz).

*4-Chloro-2-(4-chloroquinolin-3-yl)phenol* (**4i**): Yellow solid (0.28 g, 95%), ^1^H-NMR (400 MHz, CD_3_OD) δ 9.28 (s, 1H), 8.67 (d, *J* = 8.5 Hz, 1H), 8.30 (d, *J* = 8.5 Hz, 1H), 8.25 (dd, *J* = 8.5, 6.9 Hz, 1H), 8.10 (dd, *J* = 8.2, 7.2 Hz, 1H), 7.44 (d, *J* = 2.5 Hz, 1H), 7.38 (dd, *J* = 8.8, 2.6 Hz, 1H), 7.00 (d, *J* = 8.8 Hz, 1H). ^13^C-NMR (100 MHz, CD_3_OD) δ 155.3, 153.9, 147.7, 138.8, 136.8, 132.8, 132.7, 132.6, 131.8, 129.0, 127.2, 125.5, 122.8, 122.2, 118.5.

*2-(4-Chloroquinolin-3-yl)-4-methylphenol* (**4j**): Brown solid (0.25 g, 92%), ^1^H-NMR (400 MHz, CD_3_OD/CDCl_3_ = 1/4) δ 8.92 (s, 1H), 8.47 (d, *J* = 7.9 Hz, 1H), 8.24 (s, 1H), 8.01–7.88 (m, 2H), 7.13–7.08 (m, 2H), 6.87 (d, *J* = 8.0 Hz, 1H), 2.28 (s, 3H). ^13^C-NMR (100 MHz, CD_3_OD/CDCl_3_ = 1/4) δ 152.3, 148.5, 148.2, 140.6, 133.4, 132.3, 131.8, 131.2, 130.1, 129.0, 127.4, 125.5, 123.8, 120.3, 115.8, 20.0.

*2-(4-Chloroquinolin-3-yl)-4-(trifluoromethoxy)phenol* (**4k**): Yellow solid (0.29 g, 85%), ^1^H-NMR (400 MHz, CD_3_OD/CDCl_3_ = 1/4) δ 8.81 (s, 1H), 8.31 (d, *J* = 8.2 Hz, 1H), 8.16 (d, *J* = 8.2 Hz, 1H), 7.80 (dd, *J* = 8.0, 6.7 Hz, 1H), 7.69 (dd, *J* = 7.8, 7.3 Hz, 1H), 7.14–7.12 (m, 2H), 6.96 (d, *J* = 7.8 Hz, 1H). ^13^C-NMR (100 MHz, CD_3_OD/CDCl_3_ = 1/4) δ 150.9, 147.7, 142.3, 141.6, 138.9 (d, J_C-F_ = 1.6 Hz), 129.1, 127.2, 126.2, 124.5, 124.3, 122.4, 121.8, 121.0, 120.2, 119.3, 116.7, 114.4.

*2-(4-Chloroquinolin-3-yl)benzene-1,4-diol* (**4l**): Brown solid (0.19 g, 68%), ^1^H-NMR (400 MHz, DMSO*-d*_6_) δ 9.02 (s, 1H), 8.38 (d, *J* = 8.2 Hz, 1H), 8.20 (d, *J* = 8.2 Hz, 1H), 8.01 (dd, *J* = 7.5, 7.1 Hz, 1H), 7.90 (dd, *J* = 7.5, 7.4 Hz, 1H), 6.84 (d, *J* = 8.6 Hz, 1H), 6.77–6.74 (m, 1H), 6.71 (d, *J* = 2.4 Hz, 1H). ^13^C-NMR (100 MHz, DMSO*-d*_6_) δ 150.3, 149.7, 147.3, 143.5, 143.2, 131.9, 131.4, 129.3, 126.4, 126.0, 124.5, 122.0, 117.1, 117.0, 116.6.

*3-Chloro-2-(4-chloroquinolin-3-yl)phenol* (**4m**): Golden solid (0.27 g, 94%), ^1^H-NMR (400 MHz, CD_3_OD/CDCl_3_ = 2/3) δ 8.62 (s, 1H), 8.32 (d, *J* = 8.0 Hz, 1H), 8.07 (d, *J* = 8.4 Hz, 1H), 7.80–7.76 (m, 1H), 7.68–7.64 (m, 1H), 7.22 (dd, *J* = 8.2, 8.1 Hz, 1H), 7.02 (d, *J* = 7.4 Hz, 1H), 6.88 (d, *J* = 8.2 Hz, 1H). ^13^C-NMR (100 MHz, CD_3_OD/CDCl_3_ = 2/3) δ 162.3, 155.4, 150.7, 146.5, 142.2, 133.7, 129.6, 127.9, 127.7, 127.0, 125.7, 123.8, 121.5, 119.4, 113.2.

*2-(4-Chloro-5-methylquinolin-3-yl)phenol* (**4n**): Yellow solid (0.26 g, 95%), ^1^H-NMR (400 MHz, CD_3_OD/CDCl_3_ = 3/2) δ 8.65 (s, 1H), 7.95 (d, *J* = 8.3 Hz, 1H), 7.63 (dd, *J* = 8.2, 7.4 Hz, 1H), 7.46 (d, *J* = 7.3 Hz, 1H), 7.31–7.27 (m, 1H), 7.21–7.19 (m, 1H), 6.98–6.94 (m, 2H), 3.06 (s, 3H). ^13^C-NMR (100 MHz, CD_3_OD/CDCl_3_ = 3/2) δ 153.8, 148.9, 146.7, 143.3, 135.6, 132.1, 131.1, 130.2, 129.5, 129.4, 125.6 (2C), 122.5, 118.7, 115.0, 25.0.

*2-(4,6-Dichloroquinolin-3-yl)phenol* (**4o**): Orange solid (0.27 g, 94%), ^1^H-NMR (400 MHz, DMSO-*d*_6_) δ 9.83 (br s, 1H), 8.82 (s, 1H), 8.26 (d, *J* = 2.3 Hz, 1H), 8.15 (d, *J* = 8.9 Hz, 1H), 7.91 (dd, *J* = 9.0, 2.3 Hz, 1H), 7.34–7.27 (m, 2H), 7.01 (d, *J* = 7.6 Hz, 1H), 6.97–6.93 (m, 1H). ^13^C-NMR (100 MHz, DMSO-*d*_6_) δ 154.8, 152.6, 145.5, 138.8, 133.0, 131.8, 131.5, 131.0, 130.9, 130.3, 126.5, 122.8, 122.3, 119.0, 115.8.

*2-(4-Chloro-6-methylquinolin-3-yl)phenol* (**4p**): Yellow solid (0.23 g, 86%), ^1^H-NMR (400 MHz, DMSO-*d*_6_) δ 9.65 (s, 1H), 8.61 (s, 1H), 7.97 (s, 1H), 7.93 (d, *J* = 8.5 Hz, 1H), 7.63 (d, *J* = 8.4 Hz, 1H), 7.24–7.18 (m, 2H), 6.93 (d, *J* = 8.0 Hz, 1H), 6.8–6.84 (m, 1H), 2.42 (s, 3H). ^13^C-NMR (100 MHz, DMSO-*d*_6_) δ 154.9, 151.2, 145.9, 138.8, 138.0, 132.2, 131.1, 130.8, 129.9, 129.1, 125.5, 122.9, 122.6, 119.0, 115.8, 21.4.

*4-Chloro-3-(2-hydroxyphenyl)quinolin-6-ol* (**4q**): Dark-red solid (0.25 g, 91%), ^1^H-NMR (400 MHz, DMSO-*d*_6_) δ 11.03 (br s, 1H), 9.93 (br s, 1H), 9.01 (s, 1H), 8.17–8.16 (m, 1H), 7.66–7.63 (m, 2H), 7.34 (s, 2H), 7.06–6.98 (m, 2H). ^13^C-NMR (100 MHz, DMSO-*d*_6_) δ 158.8, 154.8 (2C), 144.5, 135.2, 131.5, 131.1, 131.0, 130.7, 128.2, 121.2, 119.1, 115.9, 105.5, 99.5.

*2-(4-Chloro-7-fluoroquinolin-3-yl)phenol* (**4r**): Light yellow solid (0.25 g, 92%), ^1^H-NMR (400 MHz, DMSO-*d*_6_) δ 9.78 (s, 1H), 8.81 (s, 1H), 8.37 (dd, *J* = 9.3, 6.0 Hz, 1H), 7.91 (dd, *J* = 10.0, 2.5 Hz, 1H), 7.76–7.71 (m, 1H), 7.33–7.28 (m, 2H), 7.01 (d, *J* = 8.1 Hz, 1H), 6.97–6.93 (m, 1H). ^13^C-NMR (100 MHz, DMSO-*d*_6_) δ 162.6 (d, J_C-F_ = 247.7 Hz), 154.9, 153.5, 148.2 (d, J_C-F_ = 12.8 Hz), 139.8 (d, J_C-F_ = 0.8 Hz), 131.1, 130.5 (d, J_C-F_ = 2.4 Hz), 130.1, 127.0 (d, J_C-F_ = 10.1 Hz), 122.9, 122.4, 119.0, 118.3 (d, J_C-F_ = 25.1 Hz), 115.8, 112.9 (d, J_C-F_ = 20.4 Hz).

*2-(4,7-Dichloroquinolin-3-yl)phenol* (**4s**): Orange solid (0.26 g, 90%), ^1^H-NMR (400 MHz, CD_3_OD/CDCl_3_ = 3/2) δ 8.76 (s, 1H), 8.29 (d, *J* = 9.0 Hz, 1H), 8.05 (d, *J* = 2.0 Hz, 1H), 7.64 (dd, *J* = 9.0, 2.0 Hz, 1H), 7.32–7.28 (m, 1H), 7.24 (dd, *J* = 7.4, 1.0 Hz, 1H), 6.98–6.94 (m, 2H). ^13^C-NMR (100 MHz, CD_3_OD/CDCl_3_ = 3/2) δ 156.1, 154.6, 148.8, 143.1, 137.6, 133.0, 132.6, 131.7, 130.1, 128.7, 127.6, 126.7, 123.9, 120.8, 117.1.

*2-(4-Chloro-7-methylquinolin-3-yl)phenol* (**4t**): Earthy solid (0.23 g, 86%), ^1^H-NMR (400 MHz, DMSO-*d*_6_) δ 9.79 (br s, 1H), 8.80 (s, 1H), 8.20 (d, *J* = 8.0 Hz, 1H), 7.93 (s, 1H), 7.67 (d, *J* = 8.4 Hz, 1H), 7.33–7.27 (m, 2H), 7.01 (d, *J* = 8.0 Hz, 1H), 6.95 (dd, *J* = 7.4, 7.3 Hz, 1H), 2.58 (s, 3H). ^13^C-NMR (100 MHz, DMSO-*d*_6_) δ 154.9, 151.5, 146.4, 140.9, 140.5, 131.2, 130.6, 130.2, 130.0, 127.3, 123.8 (2C), 122.5, 119.0, 115.8, 21.2.

*4-Chloro-3-(2-hydroxyphenyl)quinolin-7-ol* (**4u**): Brown solid (0.24 g, 89%), ^1^H-NMR (400 MHz, CD_3_OD) δ 8.96 (s, 1H), 8.51 (d, *J* = 9.3 Hz, 1H), 7.59 (dd, *J* = 9.4, 2.2 Hz, 1H), 7.46 (d, *J* = 2.2 Hz, 1H), 7.40–7.34 (m, 2H), 7.04–7.00 (m, 2H). ^13^C-NMR (100 MHz, CD_3_OD) δ 165.4, 156.4, 152.6, 146.7, 142.1, 132.4 (2C), 130.6, 129.2, 124.9, 123.5, 121.8, 120.8, 116.9, 103.5.

*2-(4-Chloro-8-methylquinolin-3-yl)phenol* (**4v**): Yellow solid (0.25 g, 91%), ^1^H-NMR (400 MHz, DMSO-*d*_6_) δ 9.74 (br s, 1H), 8.80 (s, 1H), 8.13 (d, *J* = 8.0 Hz, 1H), 7.73 (d, *J* = 6.8 Hz, 1H), 7.68–7.64 (m, 1H), 7.33–7.26 (m, 2H), 7.02 (d, *J* = 7.8 Hz, 1H), 6.97–6.93 (m, 1H), 2.77 (s, 3H). ^13^C-NMR (100 MHz, DMSO-*d*_6_) δ 154.9, 150.9, 146.1, 140.0, 137.0, 131.1, 130.7, 130.2, 130.0, 127.8, 125.6, 122.9, 121.9, 119.0, 115.8, 17.9.

*2-(7-Bromo-4-chloroquinolin-3-yl)benzene-1,4-diol* (**4w**): Yellow solid (0.34 g, 97%), ^1^H-NMR (400 MHz, CD_3_OD) δ 8.74 (s, 1H), 8.26–8.24 (m, 2H), 7.82 (dd, *J* = 9.0, 1.9 Hz, 1H), 6.83–6.76 (m, 2H), 6.72 (d, *J* = 2.7 Hz, 1H). ^13^C-NMR (100 MHz, CD_3_OD) δ 154.6, 151.3, 149.0 (2C), 142.7, 133.2, 132.5, 131.9, 127.4, 126.7, 125.5, 124.4, 118.3, 118.2, 117.9.

*2-(4,7-Dichloroquinolin-3-yl)benzene-1,4-diol* (**4x**): Brick-red solid (0.28 g, 91%), ^1^H-NMR (400 MHz, DMSO-*d*_6_) δ 8.83 (s, 1H), 8.31 (d, *J* = 9.0 Hz, 1H), 8.20 (d, *J* = 1.8 Hz, 1H), 7.84 (dd, *J* = 9.0, 1.8 Hz, 1H), 6.83 (d, *J* = 8.7 Hz, 1H), 6.74 (dd, *J* = 8.6, 2.7 Hz, 1H), 6.68 (d, *J* = 2.7 Hz, 1H). ^13^C-NMR (100 MHz, DMSO-*d*_6_) δ 153.3, 149.7, 147.3, 147.0, 140.0, 135.0, 131.5, 128.9, 127.6, 126.3, 124.5, 122.5, 117.0, 116.9, 116.6.

#### 3.1.3. General Procedures for the Synthesis of Benzofuro[3,2-c]Quionlines **2a**–**2x**

To a solution of the 2-(4-chloroquinolin-3-yl)phenol (1 equiv.) in DMF (5 mL), KOt-Bu (2 equiv.) was added in one portion. The reaction mixture was first stirred at 25 °C for 0.5 h, then stirred at 60 °C for 2 h. The mixture was poured into ice water. The organic phase was separated and the water phase was extracted with ethyl acetate three times (3 × 20 mL). The combined organic phases were dried over anhydrous Na_2_SO_4_. Removal of the solvent was done in vacuo, and the crude mixture was further purified using column chromatography over silica gel using petroleum ether/ethyl acetate (15:1) as eluent.

*Benzofuro[3,2-c]quinoline* (**2a**): White solid (0.19 g, 88%), m.p. 130–132 °C. ^1^H-NMR (400 MHz, CDCl_3_) δ 9.48 (s, 1H), 8.40 (d, *J* = 8.1 Hz, 1H), 8.26 (d, *J* = 8.4 Hz, 1H), 8.08 (d, *J* = 7.6 Hz, 1H), 7.80–7.73 (m, 2H), 7.69 (dd, *J* = 7.6, 7.4 Hz, 1H), 7.54 (dd, *J* = 7.8, 7.4 Hz, 1H), 7.46 (dd, *J* = 7.4, 7.3 Hz, 1H). ^13^C-NMR (100 MHz, CDCl_3_) δ 156.4, 154.9, 146.4, 143.4, 128.8, 128.2, 126.2, 126.0, 123.0, 121.6, 119.8, 119.6, 116.1, 115.3, 111.1; HRMS (ESI): calcd. for C_15_H_10_NO [M + H]^+^ 220.0757; found 220.0759.

*6-Methylbenzofuro[3,2-c]quinoline* (**2b**): White solid (0.15 g, 85%), m.p. 139–141 °C. ^1^H-NMR (400 MHz, CDCl_3_) δ 9.48–9.46 (m, 1H), 8.45 (dd, *J* = 7.8, 6.7 Hz, 1H), 8.26 (d, *J* = 8.3 Hz, 1H), 7.91 (dd, *J* = 8.6, 6.8 Hz, 1H), 7.80–7.76 (m, 1H), 7.69 (dd, *J* = 8.0, 6.9 Hz, 1H), 7.38–7.34 (m, 2H), 2.70 (s, 3H). ^13^C-NMR (100 MHz, CDCl_3_) δ 156.2, 153.9, 146.3, 143.6, 128.8, 128.1, 127.2, 125.8, 123.0, 121.4, 121.1, 119.8, 116.9, 116.2, 115.7, 14.2; HRMS (ESI): calcd. for C_16_H_12_NO [M + H]^+^ 234.0913; found 234.0906.

*6-Fluorobenzofuro[3,2-c]quinoline* (**2c**): White solid (0.17 g, 82%), m.p. 154–157 °C. ^1^H-NMR (400 MHz, CDCl_3_) δ 9.47 (s, 1H), 8.46 (d, *J* = 8.0 Hz, 1H), 8.27 (d, *J* = 8.5 Hz, 1H), 7.85 (d, *J* = 7.7 Hz, 1H), 7.84–7.79 (m, 1H), 7.72 (dd, *J* = 7.4, 7.2 Hz, 1H), 7.40 (ddd, *J* = 8.0, 7.9, 4.3 Hz, 1H), 7.31–7.27 (m, 1H). ^13^C-NMR (100 MHz, CDCl_3_) δ 156.8, 147.3 (d, *J*_C-F_ = 249.5 Hz), 146.7, 143.3, 141.7 (d, *J*_C-F_ = 11.3 Hz), 128.8, 128.7, 126.3, 125.2 (d, *J*_C-F_ = 2.6 Hz), 123.8 (d, *J*_C-F_ = 5.8 Hz), 119.8, 116.0, 115.1 (d, *J*_C-F_ = 4.0 Hz), 115.0 (d, *J*_C-F_ = 2.2 Hz), 112.7 (d, *J*_C-F_ = 16.1 Hz); HRMS (ESI): calcd. for C_15_H_9_FNO [M + H]^+^ 238.0663; found 238.0667.

*5-Methylbenzofuro[3,2-c]quinoline* (**2d**): White solid (0.17 g, 92%), m.p. 146–149 °C. ^1^H-NMR (400 MHz,CDCl_3_) δ 9.40 (s, 1H), 8.34 (d, *J* = 8.0 Hz, 1H), 8.23 (d, *J* = 8.4 Hz, 1H), 7.88 (d, *J* = 7.8 Hz, 1H), 7.74 (dd, *J* = 7.3, 7.0 Hz, 1H), 7.64 (dd, *J* = 7.8, 7.1 Hz, 1H), 7.48 (s, 1H), 7.23 (d, *J* = 7.8 Hz, 1H), 2.53 (s, 3H). ^13^C-NMR (100 MHz, CDCl_3_) δ 156.2, 155.3, 146.0, 143.1, 136.8, 128.7, 127.9, 125.8, 124.2, 119.6, 119.0, 116.1, 115.3, 111.2, 20.9; HRMS (ESI): calcd. for C_16_H_12_NO [M + H]^+^ 234.0913; found 234.0907.

*Benzofuro[3,2-c]quinolin-5-ol* (**2e**): Earthy solid (0.03 g, 12%), m.p. >250 °C. ^1^H-NMR (400 MHz, CD_3_OD/CDCl_3_ = 3/2) δ 9.30 (s, 1H), 8.33 (d, *J* = 8.1 Hz, 1H), 8.13 (d, *J* = 8.4 Hz, 1H), 7.88 (d, *J* = 8.4 Hz, 1H), 7.74 (dd, *J* = 8.3, 7.0 Hz, 1H), 7.67 (dd, *J* = 7.4, 7.3 Hz, H), 7.17 (s, 1H), 6.97 (d, *J* = 8.4 Hz, 1H). ^13^C-NMR (100 MHz, CD_3_OD/CDCl_3_ = 3/2) δ 158.0, 157.6, 157.2, 145.7, 143.3, 128.9, 128.3, 127.0, 120.8, 120.4, 117.0, 116.9, 114.2, 113.1, 98.6; HRMS (ESI): calcd. for C_15_H_10_NO_2_ [M + H]^+^ 236.0706; found 236.0707.

*5-Bromobenzofuro[3,2-c]quinoline* (**2f**): White solid (0.24 g, 85%), m.p. 195–196 °C. ^1^H-NMR (400 MHz, CDCl_3_) δ 9.46 (s, 1H), 8.40 (d, *J* = 8.0 Hz, 1H), 8.27 (d, *J* = 8.4 Hz, 1H), 7.96–7.93 (m, 2H), 7.81 (ddd, *J* = 8.3, 7.0, 1.2 Hz, 1H), 7.71 (ddd, *J* = 8.0, 7.1, 1.1 Hz, 1H), 7.61 (d, *J* = 8.2 Hz, 1H). ^13^C-NMR (100 MHz, CDCl_3_) δ 156.7, 155.1, 146.5, 143.1, 128.9, 128.6, 126.5, 126.3, 120.8, 120.5, 119.8, 119.4, 116.0, 114.7 (2C); HRMS (ESI): calcd. for C_15_H_9_BrNO [M + H]^+^ 297.9862; found 297.9869.

*5-Chlorobenzofuro[3,2-c]quinoline* (**2g**): White solid (0.17 g, 75%), m.p. 194–195 °C. ^1^H-NMR (400 MHz, CDCl_3_) δ 9.45 (s, 1H), 8.39 (d, *J* = 8.0 Hz, 1H), 8.27 (d, *J* = 8.4 Hz, 1H), 8.00 (d, *J* = 8.3 Hz, 1H), 7.81 (ddd, *J* = 8.3, 7.0, 0.4 Hz, 1H), 7.76 (s, 1H), 7.71 (ddd, *J* = 8.0, 7.1, 0.7 Hz, 1H), 7.46 (d, *J* = 8.3 Hz, 1H). ^13^C-NMR (100 MHz, CDCl_3_) δ 156.8, 155.0, 146.4, 143.1, 131.9, 128.9, 128.6, 126.2, 123.8, 120.4, 120.1, 119.7, 116.0, 114.6, 111.8; HRMS (ESI): calcd. for C_15_H_9_ClNO [M + H]^+^ 254.0367; found 254.0362.

*4-Fluorobenzofuro[3,2-c]quinoline* (**2h**): White solid (0.17 g, 89 %), m.p. 187–188 °C. ^1^H-NMR (400 MHz, CDCl_3_) δ 9.42 (s, 1H), 8.38 (d, *J* = 8.2 Hz, 1H), 8.26 (d, *J* = 8.4 Hz, 1H), 7.80 (dd, *J* = 8.2, 7.1 Hz,1H), 7.74–7.66 (m, 3H), 7.26–7.22 (m, 1H). ^13^C-NMR (100 MHz, CDCl_3_) δ 158.7 (d, *J*_C-F_ = 239.8 Hz), 157.6, 150.9, 146.5, 143.3, 128.9, 128.6, 126.2, 122.6 (d, *J*_C-F_ = 10.6 Hz), 119.8, 116.0, 115.1 (d, *J*_C-F_ = 3.7 Hz), 113.7 (d, *J*_C-F_ = 25.8 Hz), 111.9 (d, *J*_C-F_ = 9.4 Hz), 105.7 (d, *J*_C-F_ = 25.5 Hz); HRMS (ESI): calcd. for C_15_H_9_FNO [M + H]^+^ 238.0663; found 238.0664.

*4-Chlorobenzofuro[3,2-c]quinoline* (**2i**): White solid (0.21 g, 86%), m.p. 181–182 °C. ^1^H-NMR (400 MHz, CDCl_3_) δ 9.43 (s, 1H), 8.38 (d, *J* = 8.1 Hz, 1H), 8.26 (d, *J* = 8.4 Hz, 1H), 8.05 (d, *J* = 1.7 Hz, 1H), 7.80 (dd, *J* = 8.2, 7.1 Hz, 1H), 7.70 (dd, *J* = 7.7, 7.4 Hz, 1H), 7.66 (d, *J* = 8.8 Hz, 1H), 7.49 (dd, *J* = 8.8, 1.9 Hz, 1H). ^13^C-NMR (100 MHz, CDCl_3_) δ 157.2, 153.2, 146.6, 143.2, 128.9, 128.7 (2C), 126.3, 126.2, 123.1, 119.8, 119.4, 116.0, 114.5, 112.1; HRMS (ESI): calcd. for C_15_H_9_ClNO [M + H]^+^ 254.0367; found 254.0361.

*4-Methylbenzofuro[3,2-c]quinoline* (**2j**): Yellow solid (0.20 g, 94%), m.p. 149–150 °C. ^1^H-NMR (400 MHz, CDCl_3_) δ 9.45 (s, 1H), 8.40–8.38 (m, 1H), 8.25 (d, *J* = 8.4 Hz, 1H), 7.87 (s, 1H), 7.79–7.75 (m, 1H), 7.70–7.66 (m, 1H), 7.61 (d, *J* = 8.4 Hz, 1H), 7.34–7.32 (m, 1H), 2.55 (s, 3H). ^13^C-NMR (100 MHz, CDCl_3_) δ 157.7, 154.4, 147.3, 144.4, 133.8, 129.8, 129.2, 128.3, 126.9, 122.7, 120.8, 120.5, 117.3, 116.3, 111.6, 21.4; HRMS (ESI): calcd. for C_16_H_12_NO [M + H]^+^ 234.0913; found 234.0916.

*4-(Trifluoromethoxy)benzofuro[3,2-c]quinoline* (**2k**): White solid (0.16 g, 62%), m.p. 143–144 °C. ^1^H-NMR (400 MHz, CDCl_3_) δ 9.45 (s, 1H), 8.40–8.38 (m, 1H), 8.27 (d, *J* = 8.4 Hz, 1H), 7.94 (d, *J* = 1.0 Hz, 1H), 7.84–7.79 (m, 1H), 7.75–7.69 (m, 2H), 7.40 (dd, *J* = 8.9,1.6 Hz, 1H). ^13^C-NMR (100 MHz, CDCl_3_) δ 157.6, 152.8, 146.6, 144.7 (d, *J*_C-F_ = 2.1 Hz), 143.2, 129.0, 128.8, 126.3, 122.7, 119.8, 119.7, 119.6 (q, *J*_C-F_ = 255.5 Hz), 115.9, 114.8, 112.5, 112.0. ^19^F NMR (376 MHz, CDCl_3_): δ −58.1. HRMS (ESI): calcd. for C_16_H_9_F_3_NO_2_ [M + H]^+^ 304.0580; found 304.0584.

*Benzofuro[3,2-c]quinolin-4-ol* (**2l**): Light yellow solid (0.04 g, 23%), m.p. >250 °C. ^1^H-NMR (400 MHz, CD_3_OD/CDCl_3_ = 3/2) δ 9.30 (s, 1H), 8.36 (d, *J* = 7.9 Hz, 1H), 8.14 (d, *J* = 8.4 Hz, 1H), 7.76 (dd, *J* = 8.4, 8.2 Hz, 1H), 7.68 (dd, *J* = 7.7, 7.3 Hz, 1H), 7.55 (d, *J* = 8.8 Hz, 1H), 7.46 (s, 1H), 7.03 (dd, *J* = 8.9, 2.5 Hz, 1H). ^13^C-NMR (100 MHz, CD_3_OD/CDCl_3_ = 3/2) δ 157.4, 153.3, 149.5, 145.5, 143.2, 128.8, 127.6, 126.4, 122.2, 120.0, 116.4, 115.9, 115.1, 111.6, 104.7; HRMS (ESI): calcd. for C_15_H_10_NO_2_ [M + H]^+^ 236.0706; found 236.0710.

*3-Chlorobenzofuro[3,2-c]quinoline* (**2m**): White solid (0.10 g, 42%), m.p. 160–162 °C. ^1^H-NMR (400 MHz, CDCl_3_) δ 9.79 (s, 1H), 8.40–8.38 (m, 1H), 8.28 (d, *J* = 8.4 Hz, 1H), 7.83–7.79 (m, 1H), 7.72–7.68 (m, 1H), 7.65 (dd, *J* = 6.7, 2.3 Hz, 1H), 7.48–7.43 (m, 2H). ^13^C-NMR (100 MHz, CDCl_3_) δ 156.5, 155.2, 146.4, 144.5, 128.9, 128.7, 127.0, 126.6, 126.1, 123.5, 120.7, 119.8, 115.7, 114.5, 109.4; HRMS (ESI): calcd. for C_15_H_9_ClNO [M + H]^+^ 254.0367; found 254.0348.

*7-Methylbenzofuro[3,2-c]quinoline* (**2n**): Light yellow solid (0.19 g, 85%), m.p. 146–147 °C. ^1^H-NMR (400 MHz, CDCl_3_) δ 9.46 (s, 1H), 8.11–8.09 (m, 2H), 7.75 (d, *J* = 8.1 Hz, 1H), 7.64 (dd, *J* = 8.0, 7.6 Hz, 1H), 7.54 (dd, *J* = 8.2, 7.1 Hz, 1H), 7.49–7.44 (m, 2H), 3.12 (s, 3H). ^13^C-NMR (100 MHz, CDCl_3_) δ 158.4, 155.8, 148.4, 144.1, 133.6, 128.9, 128.3, 127.6, 127.0, 124.0, 122.1, 120.4, 117.3, 116.6, 112.1, 22.4; HRMS (ESI): calcd. for C_16_H_12_NO [M + H]^+^ 234.0913; found 234.0914.

*8-Chlorobenzofuro[3,2-c]quinoline* (**2o**): White solid (0.22 g, 91%), m.p. 168–169 °C. ^1^H-NMR (400 MHz, CDCl_3_) δ 9.42 (s, 1H), 8.32 (d, *J* = 2.3 Hz, 1H), 8.16 (d, *J* = 9.0 Hz, 1H), 8.07–8.05 (m, 1H), 7.72 (d, *J* = 8.2 Hz, 1H), 7.67 (dd, *J* = 9.0, 2.4 Hz, 1H), 7.57–7.53 (m, 1H), 7.47 (ddd, *J* = 7.6, 7.5, 1.0 Hz, 1H). ^13^C-NMR (100 MHz, CDCl_3_) δ 155.3, 155.0, 144.5, 143.5, 131.9, 130.4, 129.0, 126.6, 123.2, 121.3, 119.7, 118.8, 116.7, 115.9, 111.2; HRMS (ESI): calcd. for C_15_H_9_ClNO [M + H]^+^ 254.0367; found 254.0363.

*8-Methylbenzofuro[3,2-c]quinoline* (**2p**): White solid (0.17 g, 83%), m.p. 153–156 °C. ^1^H-NMR (400 MHz, CDCl_3_) δ 9.41 (s, 1H), 8.17 (br s, 1H), 8.15 (d, *J* = 8.6 Hz, 1H), 8.09–8.07 (m, 1H), 7.74 (d, *J* = 8.2 Hz, 1H), 7.61 (dd, *J* = 8.6, 1.9 Hz, 1H), 7.56–7.51 (m, 1H), 7.46 (ddd, *J* = 7.6, 7.4, 0.9 Hz, 1H), 2.63 (s, 3H). ^13^C-NMR (100 MHz, CDCl_3_) δ 156.1, 154.9, 145.0, 142.4, 136.1, 130.5, 128.5, 126.1, 122.9, 121.8, 119.6, 118.6, 116.0, 115.2, 111.0, 20.8. HRMS (ESI): calcd. for C_16_H_12_NO [M + H]^+^ 234.0913; found 234.0919.

*Benzofuro[3,2-c]quinolin-8-ol* (**2q**): Yellow solid (0.03 g, 13%), m.p. >250 °C. ^1^H-NMR (400 MHz, DMSO-*d_6_*) δ 10.41 (s, 1H), 9.44 (s, 1H), 8.30 (d, *J* = 7.6 Hz, 1H), 8.08 (d, *J* = 9.1 Hz, 1H), 7.92 (d, *J* = 8.2 Hz, 1H), 7.61 (dd, *J* = 7.9, 7.5 Hz, 1H), 7.55–7.50 (m, 2H), 7.39 (dd, *J* = 9.1, 2.3 Hz, 1H). ^13^C-NMR (100 MHz, DMSO-*d_6_*) δ 156.4, 155.5, 155.2, 142.0, 141.3, 131.3, 127.5, 124.2, 122.3, 121.6, 121.3, 117.5, 115.7, 112.1, 101.3; HRMS (ESI): calcd. for C_15_H_10_NO_2_ [M + H]^+^ 236.0706; found 236.0715.

*9-Fluorobenzofuro[3,2-c]quinoline* (**2r**): White solid (0.17 g, 77%), m.p. 150–151 °C. ^1^H-NMR (400 MHz, CDCl_3_) δ 9.47 (s, 1H), 8.39 (dd, *J* = 9.0, 6.0 Hz, 1H), 8.07 (d, *J* = 7.6 Hz, 1H), 7.89 (dd, *J* = 10.3, 2.4 Hz, 1H), 7.73 (d, *J* = 8.2 Hz, 1H), 7.54 (dd, *J* = 7.7, 7.5 Hz, 1H), 7.50–7.46 (m, 2H). ^13^C-NMR (100 MHz, CDCl_3_) δ 161.9 (d, *J*_C-F_ = 248.2 Hz), 156.5 (d, *J*_C-F_ = 1.0 Hz), 154.8, 147.4 (d, *J*_C-F_ = 12.4 Hz), 144.5, 126.3, 123.2, 121.9 (d, *J*_C-F_ = 9.8 Hz), 121.4, 119.6, 116.3 (d, *J*_C-F_ = 25.3 Hz), 115.0 (d, *J*_C-F_ = 1.9 Hz), 113.1 (d, *J*_C-F_ = 1.1 Hz), 112.9 (d, *J*_C-F_ = 20.7 Hz), 111.1; HRMS (ESI): calcd. for C_15_H_9_FNO [M + H]^+^ 238.0663; found 238.0650.

*9-Chlorobenzofuro[3,2-c]quinoline* (**2s**): Light yellow solid (0.16 g, 69%), m.p. 155–157 °C. ^1^H-NMR (400 MHz, CDCl_3_) δ 9.46 (s, 1H), 8.32 (d, *J* = 8.8 Hz, 1H), 8.24 (d, *J* = 1.9 Hz, 1H), 8.07 (d, *J* = 7.6 Hz, 1H), 7.74 (d, *J* = 8.2 Hz, 1H), 7.63 (dd, *J* = 8.8, 2.0 Hz, 1H), 7.58–7.53 (m, 1H), 7.50–7.46 (m, 1H). ^13^C-NMR (100 MHz, CDCl_3_) δ 156.2, 154.9, 146.7, 144.4, 134.2, 128.0, 126.9, 126.5, 123.2, 121.4, 121.1, 119.7, 115.6, 114.5, 111.1; HRMS (ESI): calcd. for C_15_H_9_ClNO [M + H]^+^ 254.0367; found 254.0367.

*9-Methylbenzofuro[3,2-c]quinoline* (**2t**): Yellow-green solid (0.18 g, 92%), m.p. 159–161 °C. ^1^H-NMR (400 MHz, CDCl_3_) δ 9.44 (s, 1H), 8.30 (d, *J* = 8.4 Hz, 1H), 8.07 (d, *J* = 7.1 Hz, 1H), 8.04 (br s, 1H), 7.73 (d, *J* = 8.1 Hz, 1H), 7.54–7.50 (m, 2H), 7.48–7.44 (m, 1H), 2.62 (s, 3H). ^13^C-NMR (100 MHz, CDCl_3_) δ 156.6, 154.8, 146.7, 143.3, 138.7, 128.1, 128.0, 126.0, 122.9, 121.8, 119.5, 114.7, 114.0, 111.0, 21.0; HRMS (ESI): calcd. for C_16_H_12_NO [M + H]^+^ 234.0913; found 234.0909.

*Benzofuro[3,2-c]quinolin-9-ol* (**2u**): White solid (0.03 g, 15%), m.p. >250 °C. ^1^H-NMR (400 MHz, DMSO-*d_6_*) δ 10.39 (s, 1H), 9.53 (s, 1H), 8.28–8.25 (m, 2H), 7.88 (d, *J* = 8.1 Hz, 1H), 7.58–7.54 (m, 1H), 7.50 (dd, *J* = 7.5, 6.8 Hz, 1H), 7.47 (d, *J* = 2.1 Hz, 1H), 7.34 (dd, *J* = 8.9, 2.3 Hz, 1H). ^13^C-NMR (100 MHz, DMSO-*d_6_*) δ 158.8, 157.0, 154.9, 149.0, 145.1, 126.9, 124.2, 122.4, 121.9, 120.8, 119.6, 113.7, 111.9, 111.3, 109.9; HRMS (ESI): calcd. for C_15_H_10_NO_2_ [M + H]^+^ 236.0706; found 236.0709.

*10-Methylbenzofuro[3,2-c]quinoline* (**2v**): Off-white solid (0.17 g, 79%), m.p. 203–204 °C. ^1^H-NMR (400 MHz, CDCl_3_) δ 9.49 (s, 1H), 8.27 (d, *J* = 8.0 Hz, 1H), 8.10 (d, *J* = 7.6 Hz, 1H), 7.74 (d, *J* = 8.2 Hz, 1H), 7.64 (d, *J* = 6.6 Hz, 1H), 7.60–7.52 (m, 2H), 7.48–7.45 (m, 1H), 2.91 (s, 3H). ^13^C-NMR (100 MHz, CDCl_3_) δ 156.8, 154.9, 145.4, 142.0, 136.7, 128.7, 126.1, 125.6, 122.9, 121.8, 119.6, 117.7, 116.0, 115.0, 111.0, 17.9; HRMS (ESI): calcd. for C_16_H_12_NO [M + H]^+^ 234.0913; found 234.0906.

*9-Bromobenzofuro[3,2-c]quinolin-4-ol* (**2w**): Light pink solid (0.05 g, 17%), m.p. >250 °C. ^1^H-NMR (400 MHz, DMSO-*d_6_*) δ 9.71–9.63 (m, 2H), 8.41 (s, 1H), 8.32 (d, *J* = 8.6 Hz, 1H), 7.90 (d, *J* = 8.3 Hz, 1H), 7.72 (d, *J* = 8.8 Hz, 1H), 7.62 (s, 1H), 7.06 (d, *J* = 8.4 Hz, 1H). ^13^C-NMR (100 MHz, DMSO-*d_6_*) δ 156.8, 154.6, 149.3, 147.2, 146.5, 131.5, 130.3, 122.6, 122.5 (2C), 116.8, 116.2, 115.3, 112.6, 106.1; HRMS (ESI): calcd. for C_15_H_9_BrNO_2_ [M + H]^+^ 314.9844; found 314.0560.

*9-Chlorobenzofuro[3,2-c]quinolin-4-ol* (**2x**): Pink solid (0.03 g, 12%), m.p. >250 °C. ^1^H-NMR (400 MHz, DMSO-*d_6_*) δ 9.86 (br s, 1H), 9.62 (s, 1H), 8.38 (d, *J* = 8.8 Hz, 1H), 8.23 (d, *J* = 1.9 Hz, 1H), 7.77 (dd, *J* = 8.8, 2.0 Hz, 1H), 7.71 (d, *J* = 8.9 Hz, 1H), 7.63 (d, *J* = 2.4 Hz, 1H), 7.09 (dd, *J* = 8.9, 2.4 Hz, 1H). ^13^C-NMR (100 MHz, DMSO-*d_6_*) δ 156.7, 154.7, 149.2, 147.0, 146.5, 133.8, 128.3, 127.7, 122.5, 116.7, 116.2, 115.1, 112.5, 106.1; HRMS (ESI): calcd. for C_15_H_9_ClNO_2_ [M + H]^+^ 270.0316; found 270.0318.

### 3.2. Cell Line and Culture Conditions

The cell lines used in this study were obtained from the American Type Culture Collection (ATCC). The MV-4-11 cell line and peripheral blood mononuclear cells (PBMCs) were cultured in RMPI 1640 medium (Shanghai Weike Biotechnology Co. Ltd, Shanghai, China) supplemented with 10% FBS (fatal bovine serum). All cells were grown at 37 °C in a 5% CO_2_ atmosphere.

### 3.3. Cell Viability Assay

The MV-4-11 cells and peripheral blood mononuclear cells (PBMCs) were seeded in 96-well plates at 10,000 cells per well and treated with compounds at the indicated concentrations and time intervals (3 days). The number of viable cells was assessed using CellTiter 96^®^ Aqueous Non-Radioactive Cell Proliferation Assay (Promega), according to the manufacturer’s protocol. The absorbance of the formazan at 490 nm was detected using SpectraMax^®^ 340PC384 (Molecular Device).

## 4. Conclusions

In summary, we have developed an efficient sequential chlorination/demethylation, intramolecular annulation reaction for the synthesis of a variety of substituted benzofuro[3,2-*c*]quinolines starting from 3-(2-methoxyphenyl)quinolin-4(1*H*)ones. This sequential transformation was metal-free and was conducted under mild reaction conditions. An in vitro biological evaluation indicated that these compounds showed excellent antileukemia activity and selectivity, and therefore may offer a promising hit compound for further antileukemia studies.

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
