# Peer review of "Design, Synthesis, and In Vitro Evaluation of Benzofuro[3,2-c]Quinoline Derivatives as Potential Antileukemia Agents"

_molecules, 2020, doi:10.3390/molecules25010203_

Round 1
Reviewer 1 Report
In this manuscript molecules-681655, Ying Lin et al. describe detailed synthesis of a class of benzofuro 3,2-c quiniline derivatives via a new method which the authors claim is more efficient and gives higher yields, and they further go on to evaluate these compounds for anti-leukemia activity in the AML cell line MV-4-11.
The manuscript is well-written and the description of chemistry and reaction schemes is detailed and clear. The biological evaluation is sound, and the conclusions based on their results seem reasonable. It can be accepted for publication, subject to a couple of changes.
There are a few minor corrections, as noted below:
Scheme 1: it should be one-pot demethylation/cyclization “cascade” instead of “cacade”
Section 2.2, line 123: It’s vital to “examine” cytotoxicityon normal cells (not “exam”).
Although the yields in general are high for the compounds, as claimed, yet for a number of compounds, (for instance, 2e, 2l, 2q, etc.) the yields are poor (12-23%). Since one of these compounds, compound 2e also showed a high anti-leukemic activity in the biological evaluation, it seems reasonable to expect the authors to make a comment on why they obtained a poor yield, and how they tried to, or propose to improve them.
Author Response
Responses to reviewer 1’s comments:
Comments:
Scheme 1: it should be one-pot demethylation/cyclization “cascade” instead of “cacade”.
Our response:
Thank you! The error has been corrected.
Comments:
Line 123 : It’s vital to “examine” cytotoxicity on normal cells (not “exam”)
Our response:
This error has been corrected.
Comments:
Although the yields in general are high for the compounds, as claimed, yet for a number of compounds, (for instance, 2e,2l,2q,etc.) the yields are poor (12-23%). Since one of these compounds, compound 2e also showed a high anti-leukemic activity in the biological evaluation, it seems reasonable to expect the authors to make a comment on why they obtained a poor yield, and how they tried to, or propose to improve them.
Our response:
Many thanks for the reviewer’s comments. We added a possible solution “we speculate that the low yields of the second steps may be improved by selective protection of the hydroxyl group which is not involved in the cyclization reaction” in the manuscript.

Reviewer 2 Report
P1 L17 benzofuro[3,2-c]quinilines > benzofuro[3,2-c]quinolines
P1 L20 idem
The authors mention compound 3a but there is no structure drawn, they should include it in Scheme 1
P3 L71 4a in bold
Table 1 should be in the format of row and columns rather, than the molecular drawing, this would give a more direct view of the range of yields, the authors just would have to use R1,R2, … for the different positions of substituents.
P6 L149 purified>purification
Author Response
Responses to reviewer 2’s comments:
Comments:
P1 L17 benzofuro[3,2-c]quinilines > benzofuro[3,2-c]quinolones
P1 L20 idem
Our response:
All the errors have been corrected.
Comments:
The authors mention compound 3a but there is no structure drawn, they should include it in Scheme 1.
Our response:
Thank you! The corresponding structure has been added in Scheme 1 accordingly.
Comments:
P3 L71 4a in bold.
Our response:
This error has been corrected.
Comments:
Table 1 should be in the format of row and columns rather than the molecular drawing, this would give a more direct view of the range of yields, the authors just would have to use R1, R2,….for the different positions of substituents.
Our response:
We thank this reviewer for the good suggestion. We have already made corresponding changes.
Comments:
P6 149 purified > purification.
Our response:
This error has been corrected.

Reviewer 3 Report
The manuscript of Lin et al. describes in detail a convenient synthetic method for the preparation of benzofuroquinoline derivatives, and the evaluation of their antileukemia activities. This work can be considered a follow-up of previous work by the same research group. Still, the results described in the present manuscript provide alternative synthetic methods for the target molecules, which are thoroughly described in the supplementary information and explained in detail in the main text. The scope of the method has been tested, and 24 new benzofuroquinoline derivatives have been prepared and characterized. Both the synthetic and characterization work has been competently done. The anti-leukemia activity has been also properly tested, although the impact of the results would be better evaluated by a reviewer with more solid formation on that field than myself. The manuscript is well-structured and clearly written, with a general good use of english throughout. The bibliography cited is adequate and the list contains up to date references. Therefore, I recommend acceptance with minor revisions:
Lines 43-44: it appears stated that the molecules "...showed promising antileukemic activities...". Does this correspond to previous work? If so, a reference is needed in here. If the sentence applies to the new work described in the present manuscript, this sentence should be placed at the end of the introduction section.
Scheme 1: On the scheme (upper arrow) appears "cacade". Should be "cascade"
Author Response
Responses to reviewer 3’s comments:
Comments:
Line 43-44: it appears stated that the molecules “….showed promising antileukemia activities...” Does this correspond to previous work? If so, a reference is needed in here. If the sentence applies to the new work described in the present manuscript, this sentence should be placed at the end of the introduction section.
Our response:
We thank the reviewer for the good suggestion. This sentence “…. showed promising antileukemia activities….” does not correspond to our previous work, so we have put the sentence at the end of the introduction section.
Comments:
Scheme 1: On the scheme (upper arrow) appears “cacade”. Should be “cascade”
Our response:
The corresponding change has been made accordingly.
